# Validation and Inter-rater Reliability of the Modified Videofluoroscopic Dysphagia Scale (mVDS) in Dysphagic Patients with Multiple Etiologies

**DOI:** 10.3390/jcm10132990

**Published:** 2021-07-04

**Authors:** Min Cheol Chang, Changbae Lee, Donghwi Park

**Affiliations:** 1Department of Rehabilitation medicine, Yeungnam University Hospital, Daegu 41061, Korea; wheel633@ynu.ac.kr; 2Department of Physical Medicine and Rehabilitation, Ulsan University Hospital, University of Ulsan College of Medicine, Ulsan 44033, Korea; pclucky7@gmail.com

**Keywords:** Deglutition, Deglutition disorders, Dysphagia, VDS, Reliability, Inter-rater, VFSS

## Abstract

Background: the Videofluoroscopic Dysphagia Scale (VDS) is used to interpret and predict the long-term prognosis of patients with dysphagia. However, the inter-rater agreement of the VDS was shown to be lower in a previous study. To overcome the mentioned limitation of the VDS, a modified version (mVDS) was created and applied clinically. We aimed to validate its usefulness in determining the appropriate feeding method and predicting the prognosis of dysphagia. Methods: the videofluroscopic swallowing study (VFSS) data of 50 patients with dysphagia were collected retrospectively. The VFSS data were evaluated using the mVDS, and the inter-rater reliability was calculated. We also evaluated the association between the mVDS and type of feeding method selected, and between the mVDS and presence of aspiration pneumonia in patients with dysphagia. Results: among the different parameters of mVDS, “aspiration” showed the highest reliability (k = 0.767), followed by “mastication” and “lip closure” (k = 0.648 and k = 0.634, respectively). Conversely, “triggering pharyngeal swallow” and “pyriformis residue” demonstrated the lowest reliabilities (k = 0.312 and k = 0.324, respectively). The intraclass correlation coefficient (ICC), which is used as a measure of the reliability of the total mVDS score, was 0.876. In all patients with dysphagia, the mVDS score correlated significantly with the type of feeding method selected (*p* < 0.05), and the presence of aspiration pneumonia (*p* < 0.05). Conclusion: the ICC of the total mVDS score was 0.876. Therefore, the mVDS could be a useful tool for quantifying the severity of dysphagia. It could be helpful in the analysis of the VFSS findings among patients with dysphagia in clinical settings and research.

## 1. Background

Swallowing is a complex sensorimotor process that includes a coordinated contraction and inhibition of the muscles located in the mouth, including the tongue, pharynx, larynx, and esophagus, by different levels of the central nervous system, from the cerebral cortex to the medulla oblongata [1,2,3]. In the instrumental assessment of dysphagia, several clinical tools, such as videofluoroscopic, endoscopic, manometric, and electromyographic (EMG) studies are used currently [2,3,4]. Among these, the videofluoroscopic swallowing study (VFSS) is a primary instrumental measurement tool in clinical practice [5]. It is currently thought to be the gold standard investigation used in the instrumental assessment of dysphagia [5].

For quantitative analysis of VFSS, the Videofluoroscopic Dysphagia Scale (VDS) was introduced to evaluate the severity of dysphagia and to predict the long-term prognosis in patients with dysphagia [6,7]. The VDS has 14 parameters and shows good correlation with aspiration or symptoms of penetration that develop six months after the initial onset of dysphagia [6,7]. The 14 parameters in the VDS (Table 1) represent the oral (lip closure, mastication, bolus formation, premature bolus loss, apraxia, and oral transit time) and pharyngeal (pharyngeal triggering, laryngeal elevation, epiglottic closure, pharyngeal transit time, pharyngeal coating, vallecular and pyriform sinus residues, and aspiration) functions that can be obtained from VFSS videos [6,7].

Despite the usefulness of the VDS in quantifying the severity of dysphagia, subjective bias was noted to be a limitation in a previous study [8]. The inter-rater reliability of the VDS that was assessed by 10 physicians in the previous study showed a lower rate of agreement (k < 0.20), chiefly in the following parameters: bolus formation (k = 0.153), mastication (k = 0.123), apraxia (k = 0.099), tongue to palate contact (k = 0.153), premature bolus loss (k = 0.060), and pharyngeal transit time (k = 0.165). However, the other parameters in the VDS showed a fair rate of agreement (k > 0.2, k < 0.4) [8]. This low rate of agreement suggests that there could have been subjective bias during interpretation of VDS results comprising several parameters, such as apraxia, tongue to palate contact, premature bolus loss, or bolus formation.

Therefore, in this study, we attempted to overcome the limitations of low inter-rater agreement, for which some parameters of the VDS were revised. Furthermore, the modified VDS (mVDS) was clinically applied to validate its utility. We also analyzed the association between the mVDS and the feeding methods selected based on the VFSS findings, and the presence of aspiration pneumonia.

## 2. Methods

### 2.1. Participants

Data of patients with dysphagia who underwent VFSS at Ulsan University Hospital between March 2019 and December 2020 were collected retrospectively. Data of patients who had a symptom of difficulty in swallowing were analyzed. We obtained clinical data such as age, sex, cause of dysphagia, and a history of aspiration pneumonia. The protocol for this study was approved by the Institutional Review Board of Ulsan University Hospital.

### 2.2. Criteria for Aspiration Pneumonia

A retrospective investigation was conducted to investigate the presence of aspiration pneumonia in patients with dysphagia. We reviewed the data for the presence of aspiration pneumonia 1 month before and after the VFSS [9,10]. The following data were investigated: symptoms such as fever, dyspnea, or sputum; coughing during feeding; findings on chest radiograph; hematological laboratory findings (white blood cell (WBC) counts, C-reactive protein (CRP) level, and erythrocyte sedimentation rate (ESR)); and the use of antibiotics [9,10].

It is difficult to make a definitive diagnosis of aspiration pneumonia because the diagnostic criteria is different across various studies. Thus, patients who met all the following criteria were considered to have aspiration pneumonia in this study: (1) the presence of both subjective symptoms (fever, cough, and increased purulent sputum production) and objective signs (the presence of lung infiltrates on chest radiography, coarse lung sounds, and systemic inflammation based on hematological laboratory findings such as increased CRP levels and WBC counts), (2) clinical suspicion of an aspiration event (coughing during swallowing or increased time for swallowing), and (3) no evidence of microorganisms, such as legionella or mycoplasma, which are common pathogens of atypical pneumonia [9,10]. In addition, the clinical reports from the Department of Internal Medicine were used for the diagnosis of aspiration pneumonia [9,10].

### 2.3. VFSS Protocol

This study used a video file recorded by a fluoroscopic device. During VFSS, patients were requested to swallow the following materials which had a stepwise increase in viscosity: water, nectar (51–350 cP), rice porridge (351–1750 cP), and boiled rice (greater than 1750 cP) [11]. All materials were swallowed in the sitting position and mixed with liquid barium. Dynamic fluoroscopic images were obtained in the anterior–posterior and lateral views and were recorded at the speed of 30 frames per second. The recorded VFSS images were analyzed in accordance with the penetration–aspiration scale (PAS), and patients were considered positive for aspiration if the PAS score was higher than 5 [12]. We chose the worst PAS score among the different consistencies.

All the VFSS videos were reviewed by two physicians who had at least 10 years of experience in interpreting these results. The interpreters were blinded to patient information, including sex, age, and cause of dysphagia. They analyzed the movie files on the laptop of the patients alone, described their VFSS findings, and recommended a feeding method (non-oral feeding (by Levin tube or percutaneous gastrostomy tube) vs. oral feeding with/without modification of food viscosity or texture) based on the VFSS results and clinical parameters, such as age, underlying disease, symptoms, and cognition, etc.

### 2.4. Modification of the VDS 

The mVDS was developed on the basis of the results of a previous study with low inter-rater reliability of the VDS [8]. Among the various VDS parameters, ones with a kappa value lower than 0.2 (“bolus formation,” “mastication,” “apraxia,” “tongue in palate contact,” and “pharyngeal transit time”) were revised [8]. As mentioned by previous researchers, such parameters had slightly ambiguous guidelines and multiple conceivable choices, which led to low reliability [8]. Therefore, we modified the parameters such that a binary scale could be used, or we deleted the ambiguous parameters. The mVDS is described as follows (Table 2). The parameters “bolus formation”, “apraxia”, “premature bolus loss” and “tongue to palate contact,” which had multiple conceivable choices, were deleted due to their ambiguity. The parameters “mastication” and “lip closure” were revised to include results that could be indicated in a binary scale (intact/not intact). The parameter “pharyngeal transit time” had a binary scale of response but was deleted due to low kappa values in a previous study, and because of its similarity to the “triggering pharyngeal reflex.” The parameter “laryngeal elevation” had ambiguous criteria and the kappa value was low (k = 0.202) in a previous study [8]. Thus, we revised the parameter to “laryngeal inversion,” which was reported to be an important factor in the swallowing process as per the previous study [13]. This is also because “laryngeal elevation” and “epiglottis inversion” are the results of a combination of contraction and relaxation of the suprahyoid and infrahyoid muscles.

Hence, to evaluate these VFSS findings as objective quantitative scores, the VDS (with a sum of 100 points) was modified according to the odds ratios of the various prognostic factors [6]. Therefore, after the modification of the parameters in the VDS, we standardized each score of the mVDS parameters to a total sum of 100 points (Table 2).

### 2.5. Statistical Analysis

The intra-class correlation coefficient (ICC) model 2.1 of the mVDS was calculated to test the inter-rater reliability based on the mVDS scores. The ICC model was used because it is used not only for ordinal variables but also for scale variables [14]. Ordinal variables with weighted kappa ICC values over 0.80 were considered “very good,” and those with ICC values between 0.60-0.80 were considered “good.” [8]. The consistency of the other items was evaluated using Cohen’s kappa coefficient (k). Kappa values between 0.81–0.99, 0.61–0.80, 0.41–0.60, 0.21–0.40, and 0.01–0.20 were considered to have “almost perfect agreement,” “substantial agreement,” “moderate agreement,” “fair agreement,” and “slight agreement,” respectively [8,15].

To verify the clinical usefulness of mVDS, we evaluated the associations between the total mVDS score and the various feeding methods selected, and the mVDS and the presence of aspiration pneumonia, using univariate logistic regression analysis with enter methods. To verify the clinical utility of the mVDS, two physicians discussed the differences obtained in the various parameters of the mVDS with each other and re-investigated the VFSS images before reaching a consensus. Statistical analysis was performed using the MedCalc program and the SPSS software version 22.0 (IBM, Armonk, NY, USA).

## 3. Results

### 3.1. Characteristics of Patients

The data of 50 patients (33 males and 17 females) with dysphagia were investigated. It included 7 patients with traumatic intracerebral hemorrhage (ICH) (14.0%), 7 with supratentorial stroke (14.0%), 4 with infratentorial stroke (8.0%), 3 with subarachnoid hemorrhage (SAH) (6.0%), 10 with respiratory disease (20.0%), 9 with cancer (18.0%), 3 with cervical spinal cord injury (6.0%), 2 with Parkinson disease (4.0%), 3 with dementia (6.0%), 1 with cerebral palsy (2.0%), and 1 with esophageal stricture (2.0%) (Table 3). The average age of the investigated patients was 70.86 ± 12.73 years.

### 3.2. Inter-Rater Reliability of the mVDS

The inter-rater reliability of each parameter of the mVDS is shown in Table 3. Among the various parameters of the mVDS, “aspiration” showed the highest inter-rater reliability (k = 0.767), followed by “mastication” and “lip closure” (k = 0.648 and k = 0.634, respectively). Conversely, “triggering pharyngeal swallowing” and “pyriformis residue” demonstrated low inter-rater reliabilities (k = 0.312 and k = 0.324, respectively). The ICC of the total mVDS score was 0.876 (Table 4).

### 3.3. Association between the Results of the mVDS and the VFSS, and the mVDS and the Presence of Aspiration Pneumonia

In all patients with dysphagia, the mVDS score statistically correlated with the feeding method (*p* < 0.05) and the presence of aspiration pneumonia (*p* < 0.05) (Table 5).

## 4. Discussion

Each parameter of the mVDS score demonstrated “substantial agreement” to “fair agreement” in our study (0.767–0.312 in terms of kappa value) [15]. Moreover, the total score of the mVDS demonstrated “very good” inter-rater agreement (ICC 0.876) [14]. In comparison with the inter-rater reliability of the VDS obtained in a previous study, the mVDS showed better agreement. This could be due to the modifications made to the categories that were ambiguous to score or had multiple conceivable choices.

Additionally, herein, the total score of the mVDS showed significant association with the choice of feeding method based on the VFSS results and the presence of aspiration pneumonia in patients with dysphagia, which is an important function of the VFSS. Considering these outcomes, it is possible to deduce that the mVDS can sufficiently describe and analyze the VFSS results and predict the prognosis of patients with dysphagia with relatively high reliability.

The past two decades have led to an increase in our knowledge about dysphagia and its treatment [16,17,18,19,20]. The VFSS is the most valuable and frequently used diagnostic tool for the evaluation of dysphagia [5]. The protocol for analysis of the VFSS results has been standardized for use in many studies, such as PAS [12], the modified barium swallow impairment profile (MBSImP) [21], or the national outcomes measurements system (NOMS) [22] recommended by the American speech–language hearing association. Among these protocols, the MBSImP showed a higher inter-rater reliability after a training program, but the complexity and difficulty in learning were limitations to the use of the MBSImP in real clinical settings [21]. However, the mVDS is not only relatively easy to learn and simpler than the MBSImP, but it also showed a significant association with the presence of aspiration pneumonia, which could indicate a poorer prognosis. In the mVDS, as in the VDS, a higher score indicates a greater dietary limitation and more severe dysphagia. The mVDS can be used to document (as numerical data) the swallowing function by analysis of comprehensive VFSS findings with relatively high inter-rater reliability. Therefore, the mVDS may provide more intuitive data than the conventional VFSS interpretation, which is usually focused on the presence of aspiration or penetration.

There are several limitations to our study. First, the total number of enrolled patients was relatively small, and hence, the conclusions should be interpreted with caution. Therefore, further studies with a larger number of participants are needed for better estimation of the effects. Second, inter-rater reliability was evaluated only in two physicians. These may be the reasons for the relatively high CI of the parameters of the mVDS, except “aspiration”, in our study. However, both physicians are experts in diagnosis of dysphagia and have over 10 years of experience interpreting the VFSS results. Moreover, the mVDS was fashioned by the modification of VDS parameters that were ambiguous to score or had multiple conceivable choices. Therefore, it is expected that the mVDS would show a higher reliability than the VDS. Hence, to verify these results of the mVDS, better additional inter-rater reliability studies with more physicians are necessary.

Although our research team has reported the inter-rater reliability of the mVDS and its usefulness in stroke patients with dysphagia [23], this is the first study to evaluate the inter-rater reliability of the mVDS and its usefulness by assessing the associations between the mVDS and the results of the VFSS, and between the mVDS and the presence of aspiration pneumonia in dysphagic patients with multiple etiologies. Although the inter-rater reliability of the mVDS was better than that of the VDS, it was not perfect. It could be improved by developing a standardized education program or by developing formal guidelines for interpreters. These strategies could contribute to achieving higher levels of accuracy in analysis and subsequently improving the ability to predict the prognosis of dysphagia.

## 5. Conclusions

The ICC of the total mVDS score was 0.876. Therefore, the mVDS can be a useful scale for quantifying the severity of dysphagia and it can be utilized in clinical settings and studies to measure the findings of VFSS in patients with dysphagia. However, further studies with a larger number of patients and with more interpreters are necessary for a more widespread application of the mVDS.

## Figures and Tables

**Table 1 jcm-10-02990-t001:** Videofluoroscopic dysphagia scale (VDS).

Parameter		Score	
lip closure	intact	0	4
	inadequate	2	
	none	4	
bolus formation	intact	0	6
	inadequate	3	
	none	6	
mastication	intact	0	8
	inadequate	4	
	none	8	
apraxia	none	0	4.5
	mild	1.5	
	moderate	3	
	severe	4.5	
tongue-to-palate contact	intact	0	10
	inadequate	5	
	none	10	
premature bolus loss	none	0	4.5
	<10%	1.5	
	10–50%	3	
	>50%	4.5	
oral transit time	≤1.5 s	0	3
	>1.5 s	3	
triggering of pharyngeal swallow	normal	0	
	delayed	4.5	
vallecular residue	none	0	6
	<10%	2	
	10–50%	4	
	>50%	6	
laryngeal elevation	normal	0	9
	impaired	9	
pyriform sinus residue	none	0	13.5
	<10%	4.5	
	10–50%	9	
	>50%	13.5	
coating of pharyngeal wall	no	0	9
	yes	9	
pharyngeal transit time	≤1.0 s	0	6
	>1.0 s	6	
aspiration	none	0	12
	supraglottic penetration	6	
	subglottic aspiration	12	
total score			100

**Table 2 jcm-10-02990-t002:** Modified Videofluoroscopic dysphagia scale (mVDS).

Parameters		Score
lip closure	intact/not intact	0/6
mastication	possible/not possible	0/11.5
oral transit time	≤1.5 s/>1.5 s	0/4
triggering pharyngeal swallow (swallowing reflex)	intact/delayed	0/7
epiglottis inversion	yes/no	0/13
valleculae residue	0%/<10%/≥10%, <50%/≥50%	0/3/6/9
pyriformis residue	0%/<10%/≥10%, <50%/≥50%	0/6.5/13/19.5
pharyngeal wall coating	no/yes	0/13
aspiration	intact/penetration/aspiration	0/8.5/17
total score		100

**Table 3 jcm-10-02990-t003:** Characteristics of patients with dysphagia who included in this study.

Characteristics	Mean ± Ctandard Deviation (Median; 25–75%)
age (year)	70.86 ± 12.73 (74.00; 60.0–81.25)
sex (male:female)	33 (66.0%): 17 (34.0%)
PAS grade	3.96 ± 2.77 (3.00; 2.0–7.0)
Aspiration pneumonia (yes:no)	31 (62.0%): 19 (38.0%)
Cause of dysphagia (*n*)	Traumatic ICH = 7 (14.0%)
	Supratentorial stroke = 7 (14.0%)
	Infratentorial stroke = 4 (8.0%)
	SAH = 3 (6.0%)
	Respiratory disease = 10 (20.0%)
	Cancer = 9 (18.0%)
	Dementia = 3 (6.0%)
	Cerebral palsy = 1 (2.0%)
	Esophageal stricture = 1 (2.0%)
	Parkinson’s disease = 2 (4.0%)
	Cervical spinal cord injury = 3 (6.0%)

PAS: penetration–aspiration scale, SAH: subarachonoid hemorrhage.

**Table 4 jcm-10-02990-t004:** Inter-reliability of modified videofluoroscopic dysphaga scale (mVDS).

	K	SE	95% CI	
Lip closure	0.634	0.194	0.253	1.000
mastication	0.648	0.228	0.200	1.000
oral transit time	0.363	0.17	0.029	0.696
triggering pharyngeal swallowing	0.312	0.126	0.066	0.558
epiglottis inversion	0.359	0.188	0.010	0.728
valleculae residue	0.379	0.084	0.215	0.543
pyriformis residue	0.324	0.106	0.116	0.531
pharyngeal wall coating	0.382	0.127	0.133	0.631
aspiration	0.767	0.069	0.631	0.903
	ICC		95% CI	
Total score	0.876		0.781	0.930

SE: standard error, CI: confidence interval, ICC: intra-class correlation coefficient.

**Table 5 jcm-10-02990-t005:** Univariate logistic regression analysis (with the enter method) of the association between the modified version of the Videofluoroscopic Dysphagia Scale scores and the selection of the oral feeding method, and between the modified version of the Videofluoroscopic Dysphagia Scale scores and the development of aspiration pneumonia.

	Parameter	Beta Coefficient	Standard Error	OR (95% CI)	*p*-Value
mVDS	Determining the appropriate feeding	−0.055	0.025	0.946 (0.901–0.993)	0.025
mVDS	Aspiration pneumonia	0.044	0.018	1.045 (1.009–1.083)	0.015

## Data Availability

Data available on request due to privacy/ethical restrictions.

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
