# Peer review of "Validation and Inter-rater Reliability of the Modified Videofluoroscopic Dysphagia Scale (mVDS) in Dysphagic Patients with Multiple Etiologies"

_jcm, 2021, doi:10.3390/jcm10132990_

Round 1

Reviewer 1 Report

Videofluoroscopic swallowing examination is a very crucial examination for detecting dysphagia. However, Imaging examinations of swallowing including videofluoroscopic swallowing examination and endoscopic swallowing examination are only a part of the comprehensive examination of swallowing performance and function. In general, a thorough clinical examination should precede any imaging examination.   The clinical examination can be important in tailoring specific questions to be addressed in an imaging examination and provides a comprehensive clinical profile of patients with suspected dysphagia. This study is lacking in comprehensive clinical examinations besides pneumonia related criteria. Furthermore, there is no description of postural techniques and/or swallow maneuvers adapted during videofluoroscopic swallowing examinations.  These techniques and maneuvers have a great effect on both swallowing performances and videofuluoroscopic findings.  The above-mentioned information should be added to this paper.

Author Response

Reviewer 1

Videofluoroscopic swallowing examination is a very crucial examination for detecting dysphagia. However, Imaging examinations of swallowing including videofluoroscopic swallowing examination and endoscopic swallowing examination are only a part of the comprehensive examination of swallowing performance and function. In general, a thorough clinical examination should precede any imaging examination. The clinical examination can be important in tailoring specific questions to be addressed in an imaging examination and provides a comprehensive clinical profile of patients with suspected dysphagia.

Answer: We agree with your opinion. We agreed that the imaging examination is part of looking at swallowing function and performance. However, unlike other diseases, it cannot be denied that VFSS has a really “superior status” in Dysphagia. Will it be possible to diagnose dysphagia without VFSS and determine the patient's diet or treatment methods? We can't.

We totally agree with what the reviewer says. However, this study has focused on VFSS analysis tool. Without many years of VFSS interpretation experiences, the previously existing VFSS analysis is too complex, difficult, so, it is difficult to comprehensively interpret. Therefore, in this study, we tried to analyze the effect of mVDS without loss of accuracy while analyzing VFSS more easily and conveniently in dysphagia patients.

This study is lacking in comprehensive clinical examinations besides pneumonia related criteria.

Answer: Since we retrospectively diagnosed pneumonia with the diagnostic criteria for pneumonia that were already accepted in the previous studies, we do not think this is a problem at all. We also added a reference for diagnosis of pneumonia.

Furthermore, there is no description of postural techniques and/or swallow maneuvers adapted during videofluoroscopic swallowing examinations. These techniques and maneuvers have a great effect on both swallowing performances and videofuluoroscopic findings.  The above-mentioned information should be added to this paper.

Answer: This is a study that has absolutely nothing to do with treatment methods. This study is a study on the VFSS analysis tool, which is very important in dysphagia. this study has focused on VFSS analysis tool. Without many years of VFSS interpretation experiences, the previously existing VFSS analysis is too complex, difficult, so, it is difficult to comprehensively interpret. Therefore, in this study, we tried to analyze the effect of mVDS without loss of accuracy while analyzing VFSS more easily and conveniently in dysphagia patients.

Reviewer 2 Report

The paper describe the modification of the Videofluoroscopic Dysphagia Scale and reports the data on its reliability and clinical usefulness on 50 patients with dysphagia. Overall, the study is simple and linear and the manuscript is easy to read, although the reporting must be improved.

ABSTRACT

page1 line 21: here and throughout the paper the term "correlation" should be replaced with "association". Indeed, a correlation analysis was not performed.

Conclusions of the abstract, as well as the general conclusion of the paper, should provide a sentence on the reliability results.

INTRODUCTION

The introduction should provide more context to the study. Specifically, the author should illustrate other scales available for VFSS analysis and their strenghts but also limits that make it important to revise the VDS.

Page 1 line 39 and 43: the authors define "clinical assessment" the instrumental assessment. This should be corrected.

METHODS

Page 2 line 74: remove "

How was the PAS score assigned? Was the worst PAS score among the different consistencies considered?

In the description of the modification process of the VDS no information on whats happened to the items "apraxia" and "premature bolus loss" is provided.

Table 2: massification -> mastication

Authors should provide a reference for the interpretation of the ICC as reported in the statistical analysis' paragraph.

The methods lack to describe how the feeding method was determined and which categories are considered (e.g. only oral vs non-oral?)

RESULTS

Page 5 line 163: the first sentence should be moved to the previous paragraph.

Table 4: the CIs are very large for all the items except for aspiration. This is something that should be discussed in the discussion as it has critical consequences on the possibility to reliably use the mVDS in clinical practice.

Table 5: what is the reference category for the dependent variable "feeding methods"? It is crucial to interpret the OR. Additionally, how can the authors define "adequate" the feeding method selected.

DISCUSSION

Page 7 line218: it's not true, the same authors have recently published a similar study on stroke analyzing the same outcomes of the current study in 58 patients with stroke. It should at least discussed in the light of the present results.

The research design is simple and linear, but the methods' reporting should be improved. Results are clear. Discussion

Author Response

Reviewer 2

The paper describe the modification of the Videofluoroscopic Dysphagia Scale and reports the data on its reliability and clinical usefulness on 50 patients with dysphagia. Overall, the study is simple and linear and the manuscript is easy to read, although the reporting must be improved.

ABSTRACT

page1 line 21: here and throughout the paper the term "correlation" should be replaced with "association". Indeed, a correlation analysis was not performed.

Answer: We appreciate your valuable comment. We totally agree with your comment. Following your comment, we have modified it. “correlation” -> “association”

Conclusions of the abstract, as well as the general conclusion of the paper, should provide a sentence on the reliability results.

Answer: We appreciate your valuable comment. We totally agree with your comment. Following your comment, we have added it.

INTRODUCTION

The introduction should provide more context to the study. Specifically, the author should illustrate other scales available for VFSS analysis and their strenghts but also limits that make it important to revise the VDS.

Page 1 line 39 and 43: the authors define "clinical assessment" the instrumental assessment. This should be corrected.

Answer: We appreciate your valuable comment. We totally agree with your comment. Following your comment, we have modified it. “clinical assessment” -> “instrumental assessment”
METHODS

Page 2 line 74: remove "

Answer: We appreciate your valuable comment. Following your comment, we have deleted it.

How was the PAS score assigned? Was the worst PAS score among the different consistencies considered?

Answer: We appreciate your valuable comment. You`re correct. We chose the worst PAS score among the different consistencies. We have added it in the manuscript.

In the description of the modification process of the VDS no information on whats happened to the items "apraxia" and "premature bolus loss" is provided.

Answer: We appreciate your valuable comment. Those were deleted. We have added it in the manuscript.

Table 2: massification -> mastication

Answer: We appreciate your valuable comment. You`re correct. We have modified it.

Authors should provide a reference for the interpretation of the ICC as reported in the statistical analysis' paragraph.

Answer: We appreciate your valuable comment. Following your comment, we have added a reference.

The methods lack to describe how the feeding method was determined and which categories are considered (e.g. only oral vs non-oral?)

Answer: We appreciate your valuable comment. Following your comment, we have modified it as follows;

They analyzed the movie files on the laptop of the patients alone, described their VFSS findings, and recommended a feeding method [non-oral feeding (by Levin tube or percutaneous gastrostomy tube) vs. oral feeding with/without modification of food viscosity or texture.] based on the VFSS results and clinical parameters, such as age, underlying disease, symptoms, and cognition, etc.”

RESULTS

Page 5 line 163: the first sentence should be moved to the previous paragraph.

Answer: We appreciate your valuable comment. We totally agree with your comment. Following your comment, we have modified it.

Table 4: the CIs are very large for all the items except for aspiration. This is something that should be discussed in the discussion as it has critical consequences on the possibility to reliably use the mVDS in clinical practice.

Answer: We appreciate your valuable comment. We totally agree with your comment. We have added it in the limitation part of discussion section. Thank you.

Table 5: what is the reference category for the dependent variable "feeding methods"? It is crucial to interpret the OR. Additionally, how can the authors define "adequate" the feeding method selected.

Answer: We appreciate your valuable comment. We totally agree with your comment. Following your comment, we have added it in the method section.

They analyzed the movie files on the laptop of the patients alone, described their VFSS findings, and recommended a feeding method [non-oral feeding (by Levin tube or percutaneous gastrostomy tube) vs. oral feeding with/without modification of food viscosity or texture.] based on the VFSS results and clinical parameters, such as age, underlying disease, symptoms, and cognition, etc.”

DISCUSSION

Page 7 line218: it's not true, the same authors have recently published a similar study on stroke analyzing the same outcomes of the current study in 58 patients with stroke. It should at least discussed in the light of the present results.

Answer: We appreciate your valuable comment. Following your comment, we have updated a recent research. Thank you.

The research design is simple and linear, but the methods' reporting should be improved. Results are clear. Discussion